# Comparison of a VR Stylus with a Controller, Hand Tracking, and a Mouse for Object Manipulation and Medical Marking Tasks in Virtual Reality

**Hanna-Riikka Rantamaa [1,†], Jari Kangas [1,†] , Sriram Kishore Kumar [1], Helena Mehtonen [2], Jorma Järnstedt [2] and Roope Raisamo [1,*]**

1   Faculty of Information Technology and Communication Sciences, Tampere University,
    33014 Tampere, Finland
2   Medical Imaging Centre, Department of Radiology, Tampere University Hospital, 33520 Tampere, Finland
*   Correspondence: roope.raisamo@tuni.fi
†   These authors contributed equally to this work.

**Abstract:** In medical surgery planning, virtual reality (VR) provides a working environment, where 3D images of the operation area can be utilized. VR allows 3D imaging data to be viewed in a more realistic 3D environment, reducing perceptual problems and increasing spatial understanding. In the present experiment, we compared a mouse, hand tracking, and a combination of a VR stylus and a grab-enabled VR controller as interaction methods in VR. The purpose was to investigate the suitability of the methods in VR for object manipulation and marking tasks in medical surgery planning. The tasks required interaction with 3D objects and high accuracy in the creation of landmarks. The combination of stylus and controller was the most preferred interaction method. According to subjective results, it was considered as the most appropriate because it allows the manipulation of objects in a way that is similar to the use of bare hands. In the objective results, the mouse interaction method was the most accurate.

**Keywords:** 3D visualization; virtual reality; hand tracking; controller interaction; mouse interaction; 3D imaging

## 1. Introduction

Virtual reality (VR) makes it possible to create computer-generated environments that replace the real world. For example, the user can interact with virtual object models more flexibly using various interaction methods than with real objects in the real environment. VR has become a standard technology in research, but it has not been fully exploited in professional use, even if its potential has been demonstrated.

In the field of medicine, X-ray imaging is routinely used to diagnose diseases and anatomical changes as well as for scientific surveys [1]. In many cases, 2D medical images are satisfactory, but they can be complemented with 3D images for more complex operations where a detailed understanding of the 3D structures is needed.

When planning surgeries, medical doctors, surgeons, and radiologists study 3D images. Viewing the 3D images in 2D displays can present issues to control object position, orientation, and scaling. Using VR devices, such as head-mounted displays (HMDs), 3D images can be more easily perceived when viewed and interacted with in a 3D environment than with a 2D display. Understanding the 3D images correctly is important for accurate work. For the medical professionals to be able to perform the same tasks in VR as they perform in 2D, the interaction methods need to be studied properly. The interaction method needs to be accurate, reasonable, and suitable for the medical tasks. Because we talk about medical work, the accuracy is crucial to avoid as many mistakes as possible. König et al. [2] studied an adaptive pointing for the accuracy problems caused by hand

tremor when pointing at distant objects. Kumar et al. [3] studied three different alternative hand interaction methods for plane manipulation in VR to compensate the hand tracking inaccuracy. The used interaction method also needs to be natural so that the doctors will easily use it in their daily work and can still focus on their primary tasks without paying too much attention to the interaction method. Kangas et al. [4] compared three combinations of hand- and controller-based interaction methods and showed that there is a compromise between the accuracy and naturalness (natural, described by Navarro and Sundstedt [5] as an interaction concept of "using your own body"). One typical task for the doctors is selecting (marking) landmark locations in anatomical structures and areas on the surface of the 3D object. The selected points create the operative area, or they can be used for training.

For 2D content, a mouse is one of the best options for interaction due to its capability to point at small targets with high accuracy and the fact that many users are already experienced with this device [6]. Mouse cursors can be used for 3D pointing with ray-casting [7], which allows for the pointing of distant objects as well. The familiarity and accuracy make the mouse a worthy input method in VR, even though it is not a 3D input device. Six DoF (degrees of freedom) controllers have been identified as an accurate interaction method [8,9] and they are typically used in VR environments [10]. Controllers enable direct manipulation, and the reach of distant objects is different than with the mouse with ray-casting. Other devices, such as styluses, have been studied in pointing tasks previously [6,11]. As the controller provided good interaction capability and the stylus provided a pointing capability, we investigated the performance of them together in selected tasks.

The cameras and sensors on HMD devices also allow hand tracking without hand-held input devices. Pointing at objects with a finger is a natural act for humans, so hand interaction can be expected to be well received. While hand interaction has compared unfavorably against the controller for accurate object manipulation [9,12], we selected hand interaction as one of the conditions because of the high expectations of naturalness and easiness expressed in our interviews of medical professionals.

There are other possible interaction methods that could be used. For example, a three-dimensional mouse in VR would have been a possible option. That is a less common device, though, and we did not want to introduce that to our participants. We selected these three interaction methods mainly as they are common methods and easily available, but also because of some specific interests. The bare hands method was seen as interesting because, if that would work reliably and accurately, the user would not need to hold any extra devices. The mouse method would be familiar to most current users of IT systems. The VR controller and stylus are common devices for VR systems, and we wanted to compare them (as common alternatives) against the other methods.

We decided to use a landmark definition task to assess the three interaction conditions. The conditions were a standard mouse, bare hands, and a handheld controller with a VR stylus. The selected interaction devices have been studied earlier in different applications, but they have not been compared for the accuracy and easiness in the same task. The novelty of the current experiment is in studying these three interaction methods in a single application and measuring the participant evaluations in many usage-related attributes.

All methods were used in a VR environment to minimize additional variation between methods and to focus the comparison on interaction techniques. The use of the HMD also allowed the participants to easily study the target from different directions by moving their head. In the medical marking task, the doctor observes the anatomical structures by turning and moving the 3D object and at the same time looking for the best location for the landmark. The main difference to the general marking task is the need for high accuracy in defining the medical marking location. The time spent for the manipulation is not easily separated from the time spent in the final marking. The doctor decides during the manipulation from which angle and how the marking will be performed, which will affect the marking time. This made the application of Fitts' law [13] impossible in our study, as it requires that a participant cannot influence target locations.



We had 12 participants who were asked to perform simplified medical surgery marking tasks. To study the accuracy of the interaction methods, we created an experiment where in the 3D object there was a predefined target that was marked (pointed+selected). In the real medical case, the doctor would define the target, but then the accuracy could be easily measured. This study focused on subjective evaluations of interaction methods, but also included objective measurements.

The main contributions of this experiment are in showing the following findings in the marking task: (1) The handheld controller enabled a natural interaction for manipulation resembling the use of bare hands and was approved by the participants. (2) The stylus allowed more consistent marking accuracy compared to the bare hands use. (3) Using a mouse, the participants were able to achieve higher accuracy than with the bare hands method.

This paper is organized as follows: First, we go through the background of object manipulation and marking, interaction methods in a 3D environment, and jaw osteotomy surgery planning (Section 2). Then, we introduce the compared interaction methods and used measurements (Section 3), as well as go through the experiment (Section 4) including the apparatus, participants, and study task. In the end, the results are presented (Section 5) and discussed (Section 6).

## 2. Background

In this section, we go through the background of the task and the usual input devices that are used in object manipulation and marking tasks.

### 2.1. Object Manipulation and Marking

Object manipulation, i.e., rotating and translating the object in 3D space, and object marking, i.e., putting a small landmark on the surface of an object, have been used as separate tasks when different VR interaction methods have been studied. Sun et al. [14] studied a 3D positioning task that involved object manipulation. When a mouse and a controller were compared for precise 3D positioning the mouse was found as the more precise input device. Pham and Stuerzlinger studied object selection without manipulation in [6] and demonstrated that a 3D pen outperformed controllers. Participants also liked the 3D pen more than the controller. Allgaier et al. [15] compared different input devices for two medical tasks, liver surgical planning and craniotomy training. Their results show that participants preferred the VR Ink, followed by the controller. The data gloves and craniotome device were least liked. Argelaguet and Andujar [16] studied 3D object selection techniques in VR and Dang [17] studied 3D pointing techniques. As there were no clear standard techniques for 3D object selection or 3D pointing, Argelaguet and Andujar [16] and Dang [17] both introduced new practices in studying new techniques in 3D UIs. Kangas et al. [18] compared a mouse and two haptic interaction methods for a landmark selection task in VR demonstrating that expert users preferred the haptic methods because of naturalness.

In earlier work using bimanual techniques, Balakrishnan and Kurtenbach [19] presented a study where dominant and non-dominant hands had their own tasks in a virtual 3D scene. The bimanual technique was found faster and preferable. People typically use both their hands to cooperatively perform the most skilled tasks [19,20] where the dominant hand is used for the more accurate functions, and the non-dominant hand determines the context such as holding a canvas when the dominant hand is used to draw. The result is optimal when bimanual techniques are designed by utilizing the strengths of both dominant and non-dominant hands.

### 2.2. Input Devices for Object Manipulation and Marking
### 2.2.1. Mouse

A mouse is a common, familiar, and accurate device for 2D content to point at small targets with high accuracy [6]. The mouse is also a common device to perform medical surgery planning [10]. Many studies have used a mouse cursor for 3D pointing with ray-



casting [6,7,10,21,22]. The ray-casting technique is easily understood, and it is a solution for reaching objects at a distance [23].

Compared to other interaction methods in VR, the issue of the discrepancy between the 2D mouse and a 3D environment has been reported [16], and manipulation in 3D requires a way to switch between dimensions [24]. Balakrishnan et al. presented Rocking'Mouse to select in a 3D environment while avoiding hand fatigue. Kim and Choi [25] mentioned that the discrepancy creates a low user immersion. In addition, the use of a mouse usually forces the user to sit down next to a table instead of standing. The user can rest their arms on the table while interacting with the mouse, which decreases hand fatigue. Johnson et al. [26] stated that fatigue with mouse interaction will appear only after 3 h.

Bachmann et al. [27] found that the Leap Motion controller has a higher error rate and higher movement time than the mouse. Kim and Choi [25] showed in their study that the 2D mouse has a high performance in working time, accuracy, ease of learning, and ease of use in VR. Both Bachmann et al. and Kim and Choi found the mouse to be accurate but Li et al. [10] pointed out that with difficult marking tasks a small displacement of a physical mouse would lead to a large displacement on the 3D object in the 3D environment.

### 2.2.2. Hands

Hand interaction is a common VR interaction method. Voigt-Antons et al. [28] compared free hand interaction and controller interaction with different visualizations. Huang et al. [8] compared different interaction combinations between free hands and controllers. Both found that hand interaction has lower precision than the controller interaction, due to hand tracking performance limitations. With alternative solutions, such as a Leap Motion controller [29,30] or using wearable gloves [31], the hand interaction can be performed more accurately. Physical hand movements create a natural and realistic experience of interaction [8,32]; therefore, hand interaction is still an active area of research.

### 2.2.3. Controllers

Six DoF controllers are the leading control inputs for VR [8]. When using controllers as the interaction method, marking and selecting are usually performed using the triggers or buttons on the controller. Handheld controllers are described as stable and accurate devices [8,9]. However, holding extra devices in hands may become inconvenient if the hands are needed for other tasks between different actions. When interacting with hands or controllers in VR, hand fatigue is one of the main issues [16,33].

### 2.2.4. VR Stylus

A VR stylus is a pen-like handheld device that is used in VR environments as a controller. The physical appearance of the Logitech VR Ink stylus ("VR Ink Pilot Edition", Logitech, 28 April 2021, https://www.logitech.com/en-roeu/promo/vr-ink.html) is close to a regular pen except it has buttons that enable different interactions, e.g., selecting, in VR. Batmaz et al. [34] have studied the Logitech VR Ink stylus for selection in virtual reality. They found that using a precision grip there are no statistical differences on the marking if the distance of the target is changing. Wacker et al. [11] presented a design for a VR stylus that can point and select in mid-air at the touch of a button. For object selection, the users preferred a 3D pen over a controller in VR [6].

### *2.3. Jaw Osteotomy Surgery Planning*

Cone beam computed tomography (CBCT) is a medical imaging technique that produce 3D images that can be used in virtual surgery planning. Compared to previous techniques that were used in medical surgery planning, such as cast models, virtual planning with CBCT images has extra costs and time requirements [35]. However, the technique offers several advantages for planning accuracy and reliability [1]. CBCT images can be used as 3D objects in VR for surgery planning with an excellent match to real objects [35].

Ayoub and Pulijala [36] reviewed different studies about virtual and augmented reality applications in oral and maxillofacial surgeries.

In virtual surgery planning, the procedures for surgery are implemented and planned beforehand. The real surgery is performed based on the virtual plan. Common tasks in dental planning are specifying the location of impacted teeth, preventing nerve injuries, or preparing guiding flanges [1]. In VR, these can be performed by marking critical areas or drawing cutting lines on to the objects. Virtual planning can be used in student education as well, where the procedures can be realistically practiced. Reymus et al. [37] found that students understood the mouth anatomy better after studying 3D objects in a VR environment than from regular 2D images. The objects can be closer, bigger, and move in the depth direction in a 3D environment compared to a 2D environment [38].

Tasks, such as understanding the 3D object and marking critical areas on it, need to be performed in medical surgery planning. However, working with 3D objects in a 2D environment makes the task more difficult. Hinckley et al. [39] studied issues for developing effective free-space 3D user interfaces. Appropriate interaction and marking methods help to understand 3D objects and perform the required tasks in VR. In this study, we evaluated three methods for VR object manipulation and marking and examined the performances in simplified medical surgery planning tasks.

## 3. Material and Method

For an interaction methods comparison, we created an experiment where a participant was asked to study and mark (point+select) predefined targets on 3D objects. In this section, we first describe the used interaction methods that we are comparing. After that, we describe the used measurements.

### 3.1. Mouse

In the first interaction method, a regular mouse was used inside a VR environment (Figure 1). In the VR environment, there was a visualized mouse model that the participant was able to move by manipulating the physical mouse and controlling the direction of a ray starting from the model. The ray was always visible in the mouse interaction.

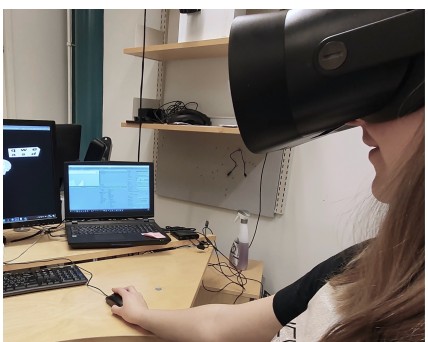 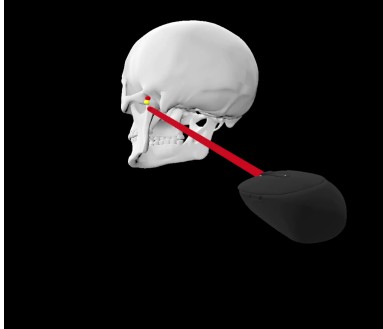

**Figure 1.** Mouse interaction method outside VR (**left**). Mouse marking method inside VR and the study task (**right**).

The mouse was used one-handed when the other two methods were two-handed. The mouse was also used to perform two functions, manipulation and marking, while these functions had been separated in other methods into different hands. In addition, the mouse used ray-casting, ray from the mouse, while the two other methods did not use it. The other methods used direct mid-air object manipulation.

The participant could rotate the object in 3 dimensions using the mouse movements with a right click. For the 3D translations, the participant used the scroll button. Using the scroll wheel, the user can zoom in and out (translate in z-axis) and, when the user presses the scroll button and moves the mouse, the user can translate up–down and sideways

(translate in x- and y-axes). Landmarks were made by pointing at the target with the ray and pressing the left button.

For the real-world mouse to be visible inside VR, pass through is not really required even though the mouse was visible in our study. After wearing the headset, the user could see the virtual mouse that was positioned to where the physical mouse was located. When the user moved the physical mouse sideways, the movement was converted to a horizontal rotation of the beam from the virtual mouse, and, when the mouse was moved back and forth, the movement was converted to a vertical rotation of the beam. The beam stops at the first object surface, where the marking would happen. This way the user could cover a large space like using a mouse in 2D displays. To improve ergonomics, the user could configure the desk and chair for their comfort.

### 3.2. Hands

As the second interaction method, the participant used bare hands. The left hand was for object manipulation and the right hand for object marking. The participant could pick up the 3D object by a pinch gesture with their left hand, to rotate and move the object. Marking was performed with a virtual pen. In the VR environment, the participant had the virtual pen attached to their right palm, near to the index finger (Figure 2, right). As the palm was moved, the pen moved accordingly. When the virtual pen tip was close to the target, the tip changed its color to green to show that the pen was touching the surface of the object. The landmark was put on the surface by bending the index finger and pressing the pen's virtual button. The participant had to keep their palm steady when pressing the button to prevent the pen from moving.

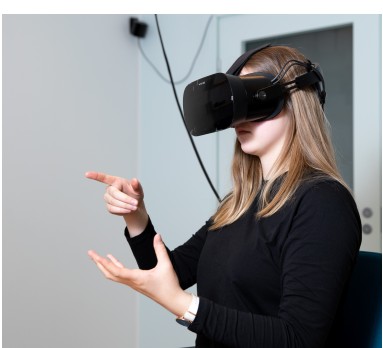 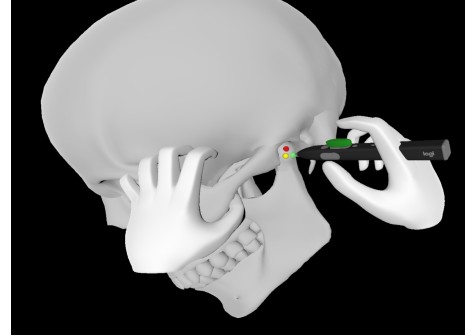

**Figure 2.** Hands interaction method outside VR (**left**). Hands marking method inside VR and the study task (**right**).

### 3.3. Controller and VR Stylus

The third interaction method was based on having a controller on participant's left hand for the object manipulation and a VR stylus on the right hand for the marking (Figure 3). The participant grabbed the 3D object with a hand grab gesture around the controller to rotate and move the object. The markings were made with the physical VR stylus. The VR stylus was visualized in VR as was the mouse, so that the participant knew where the device was located. The participant pointed at the target with the stylus and pressed its physical button to make the landmark. The act of press was identical to the virtual pen press in the hands method. There was passive haptic feedback when touching the physical VR stylus, which did not happen with the virtual pen.

There have been some supporting results for using mice in VR [10,23,25,27] but the 2D mouse is not fully compatible with the 3D environment [25]. We studied the ray method with the mouse to compare it against hands and controller+stylus for 3D object marking. We also compared hands without any devices to a method with a device in one or two hands. The marking gesture was designed to be similar in the hands and controller+stylus methods to be able to compare the effect of the devices.

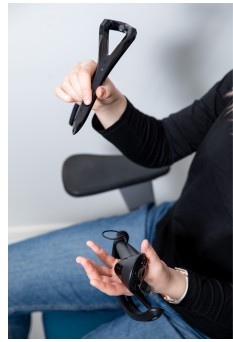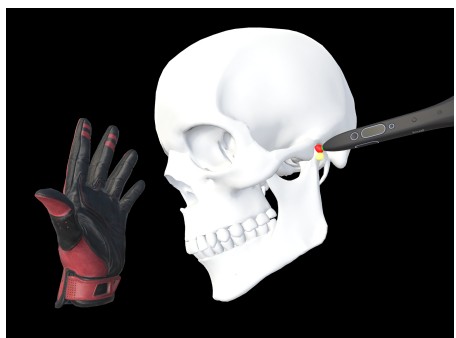

**Figure 3.** Controller interaction method outside VR (**left**). Stylus marking method inside VR and the study task (**right**).

### 3.4. Measurements and the Pilot Study

The participant was asked to make a marking as close to the target location as possible. We prepared five different objects for the experiment. We used Euclidean distance to measure the distance between the target and the participant's marking. The trial completion times were measured from when the first object appeared until the participant was ready and pressed a "Done" button for the last, the fifth object. The participant was able to remark the target if s/he was dissatisfied with the current marking. We also counted how many markings were made to see if any of the interaction methods required more remarking than the other methods.

A satisfaction questionnaire was filled after each interaction method trial. There were a question and seven satisfaction statements that were evaluated on a Likert scale from 1 (strongly disagree) to 5 (strongly agree). The statements were grouped so that the question and the first statement were about the overall feeling and the rest of the statements were about object manipulation and landmark creation separately. The statements were as follows:

- Would you think to use this method daily?
- Your hands are NOT tired.
- It was natural to perform the given tasks with this interaction method.
- It was easy to handle the 3D objects with this interaction method.
- The interaction method was accurate.
- The marking method was natural.
- It was easy to make the marking with this marking method.
- The marking method was accurate.

The statements were designed to measure fatigue, naturalness, and accuracy as they have been measured in earlier studies [8,16,32] as well. Accuracy was also measured from data to see if the objective and subjective results are consistent. With these statements, it was possible to measure the easiness and ability to use the method daily unlike from objective data.

In the questionnaire, there were also open-ended questions about positive and negative aspects of the interaction method. In the end, the participant was asked to rank the interaction methods in order from the most liked to the least liked.

A pilot study was arranged to ensure that the tasks and the study procedure were feasible. Based on the findings in the pilot study, we modified the introduction to be more specific and added a mention about the measured features. We also added the ability to rotate the 3D object even after the mouse ray moved out of the object. The speed of the mouse ray in the VR environment was increased to better match the movements of the real mouse.

### 3.5. Statistical Measures

We used two different statistical tests to analyze possible statistically significant differences between different parameter sets. For objective data (completion times, number

of markings, and accuracy) we used the paired *t*-test. For all attributes, we examined that the dependent variables are continuous, the observations are independent of one another, the dependent variables are roughly normally distributed, and the dependent variables do not contain outliers.

For data from evaluation questionnaires (fatigue, daily use, naturalness, easiness, and subjective accuracy), we first used the Friedman test to see if any statistically significant differences appeared, and then we used the Wilcoxon signed rank test as it does not assume the numbers to be in ratio scale or to have a normal distribution. For the Friedman test, we checked that the data were always a random sample from the population, there was no interaction between conditions, the data were measured on three or more different occasions, and the data were ordinal.

The study software saved the resolution of time in milliseconds and the resolution of distances in meters. To clarify the analysis, we converted these into seconds and millimeters.

## 4. Experiment

In this section, we describe the experiment arrangements. First, we describe the participant selection and participants, then the apparatus and lastly the experiment procedure.

### 4.1. Participants

We recruited 12 participants for the study. The number of participants was decided based on a power analysis for a paired *t*-test and the Wilcoxon signed rank test, assuming a large effect size,  power level of 0.8, and alpha level of 0.05. The post hoc calculated effect sizes (Cohen's *d* or *R* value, for paired *t*-test or Wilcoxon signed rank test, respectively) are reported together with the *p*-values in Results, Section 5 for comparison to the assumption of a large effect size. Ten of the participants were university students and two were full-time employees, in fields not related to medicine or dentistry. The ages varied from 21 to 30 years. The mean age was 25 years. There were 6 female participants and 6 male participants. Earlier VR experience was asked on a scale from 0 to 5, and the mean was 1.75. Two participants did not have any earlier experience. One participant was left-handed but was used to using mice with the right hand. Other participants were right-handed.

### 4.2. Apparatus
#### 4.2.1. Software, Hardware, and Hand Tracking

The experiment software was built using the Unity software ("Unity Real-Time Development Platform", Unity, 28 April 2021, https://unity.com/). With all methods, we used the Varjo VR2 Pro headset ("Varjo VR-2 Pro", Varjo, 28 April 2021, https://varjo.com/products/vr-2-pro/), which has an integrated vision-based hand tracking system that was used for hands interaction. Hands were tracked by an Ultraleap Stereo IR 170 sensor mounted on a Varjo VR2 Pro. For the controller+stylus, we used a Valve Index Controller ("The Valve Index controller", Valve, 28 April 2021, https://www.valvesoftware.com/en/index/controllers/) together with a Logitech VR Ink stylus. These were tracked by SteamVR 2.0 base stations ("SteamVR Base Station 2.0", Vive, 15 December 2021, https://www.vive.com/eu/accessory/base-station2/) around the experiment area.

#### 4.2.2. Object Manipulation and Object Marking

The study task combined two phases: an object manipulation phase where the object was rotated and translated in 3D space and an object marking phase where a small landmark was put on the surface of an object. In the object manipulation phase, the participant either selected the 3D object by mouse ray or pinched or grabbed the 3D object with a hand gesture. The 3D objects did not have any physics and laid in mid-air. By rotating and translating the object, the participant could view the object from different angles. The participant could also use head moves to change their point-of-view.

Instead of only pointing at the target, the marking needs to be confirmed. This allowed us to measure the marking accuracy and if the user understood the 3D target's location

relative to the pointing device. The participant could either release the 3D object in mid-air or hold it in their hand when hands or controller+stylus were used in the object marking task. The marking was performed either by pointing by mouse ray and clicking with left click, touching the target with the virtual pen and marking it with a hand gesture, or touching and marking with the VR stylus.

*4.3. Procedure*

First, the participant was introduced to the study, and s/he was asked to read and sign a consent form and fill in a background information form. For all conditions, the facilitator would demonstrate him/herself the system functions and the controls. Each participant had an opportunity to practice before every condition. The practice task was to move and rotate a cube having several target spheres and to mark those targets as many times as needed to get to know both the interaction and the marking methods. After the participant felt confident with the used method, s/he was asked to press the Done button, and the real study task appeared.

The participant was asked to find and mark a target on the surface of each 3D object. The target was visible all the time whereas the participant's marking was created by the participant. When the target was found, it was first pointed at and then marked. The aim was to place the participant's landmark (a yellow sphere) inside the target sphere (red) (see Figures 1–3 right). Each 3D object had one target on it and the task was repeated five times per each condition. I.e., the participants marked one target on five different objects, which created one trial. The order of 3D objects was the same to all participants: lower jaw, heart, skull, tooth, and skull. The order of the interaction methods was counter-balanced between the participants using balanced Latin Squares to compensate possible learning effects. The target locations on the 3D object (three locations on each object, one of which was used for each trial, respectively) were predefined and presented in the same order for the participants. The target locations were counter-balanced between the conditions.

The used task needed both object manipulation (rotating and translating) and marking (pointing and selecting). By combining the manipulation and marking tasks together, we created a task that simulates a task that medical professionals would perform during virtual surgery planning. Both the object manipulation and marking are needed by the medical professionals. The marking is relevant when selecting specific locations and areas of a 3D object and it requires accuracy to mark the landmarks in relevant locations. This medical marking task does not differ from regular marking tasks in other contexts as such, but the accuracy requirements are higher. By manipulating the 3D object, the professional has an option to look at the pointed area from different angles to verify its specific location in 3D environment.

A satisfaction questionnaire was filled in after each interaction method trial, and, after all three trials, a questionnaire was used to rank the conditions.

**5. Results**

In this section, we report the findings of the study. First, we present the objective results from data collected during the experiment, and then the subjective results from the questionnaires.

*5.1. Objective Results*

The trial completion times (Figure 4, top left) include both object manipulation and marking. The results had some variation, but the distributions of the median values for each of the interaction methods were similar and there were no significant differences between the mouse and hands methods ($t = -1.50, p = 0.16$), between the mouse and controller+stylus methods ($t = 1.28, p = 0.22$), or between the hands and controller+stylus methods ($t = 1.58, p = 0.14$). The degrees of freedom was 11 in all pairs. The completion time varied slightly depending on how much VR experience the participant had before, but there were no statistically significant differences.

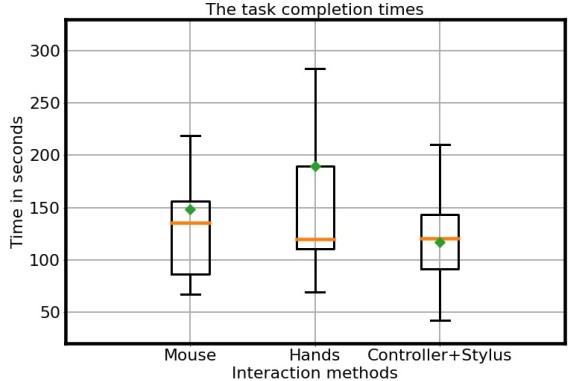

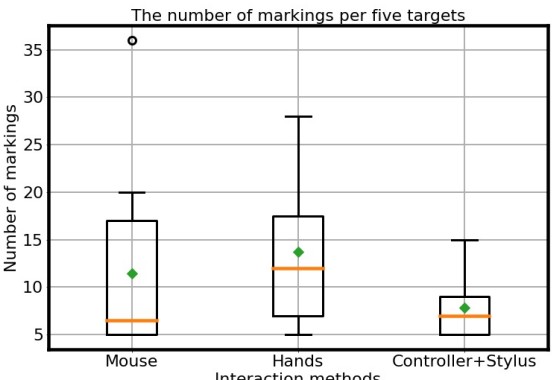

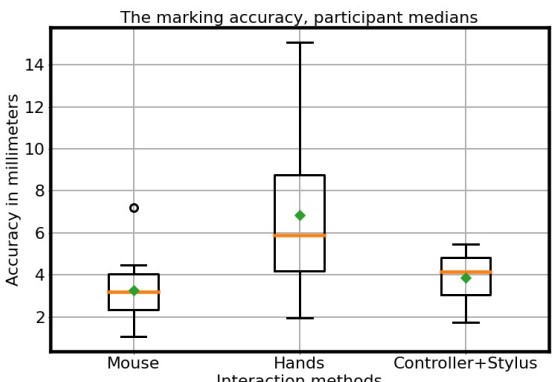

**Figure 4.** Box plot visualizations of the objective measurements. The trial completion times for different conditions (**top left**): The median values for each participant are rather similar between the methods. There were two outlier values (by the same participant, for mouse and hands conditions) that were removed from the visualization. The number of markings per five targets (**top right**): There were some differences between the interaction methods (the median value for hands was higher than for the other methods), but no significant differences. The marking accuracy (**bottom**): There was a significant difference between mouse and hands conditions, but no other significant differences.

The number of markings performed before the task completion varied between the interaction methods (Figure 4, top right). The median values for the mouse, hands, and controller+stylus conditions were 6.5, 12, and 7 markings, respectively. However, there were no statistically significant differences between the mouse and hands methods ($t = -1.02$, $p = 0.33$), between the mouse and controller+stylus methods ($t = 1.30$, $p = 0.22$), or between the hands and controller+stylus methods ($t = 2.32$, $p = 0.04$). The degrees of freedom was 11 in all pairs. Some participants performed many markings at a fast pace (2–3 markings per second) leading to a high number of total markings.

There were some clear differences in the final marking accuracy between the interaction methods (Figure 4, bottom). The median values for the mouse, hands, and controller+stylus methods were 3.2, 5.9, and 4.2 mm, respectively. The variability between participants was highest with the hands method. We found a statistically significant difference between the mouse and hands methods ($t = -3.68$, $p = 0.004$, Cohen's $d = 1.178$ (Cohen's $d \geq 0.8$ is considered a large effect size)) using a paired $t$-test and Bonferroni-corrected $p$-value limit 0.017 (=0.05/3). The degrees of freedom was 11 in all pairs. There were no statistically significant differences between the mouse and controller+stylus methods ($t = 0.98$, $p = 0.35$) or hands and controller+stylus methods ($t = 2.70$, $p = 0.021$).

### 5.2. Subjective Data

Friedman tests showed statistically significant differences in daily use ($Chi = 14.3$, $p = 0.002$), interaction naturalness ($Chi = 17.0$, $p < 0.001$), interaction easiness ($Chi = 22.0$, $p < 0.001$), interaction accuracy ($Chi = 12.2$, $p = 0.007$), marking easiness ($Chi = 8.2$, $p = 0.039$), and ranking ($Chi = 16.2$, $p < 0.001$). The degrees of freedom was 11 in all cases. There were no significant differences in marking naturalness ($Chi = 4.8$, $p = 0.09$) or marking accuracy ($Chi = 5.8$, $p = 0.053$). In evaluations for tiredness, there were no significant differences ($Chi = 2.96$, $p = 0.23$, Figure 5, left). Most participants did not feel tired using any of the methods, but the experiment was rather short.

In pairwise tests of everyday use using the Wilcoxon signed rank test we found significant differences (Figure 5, right). We found statistically significant differences between the mouse and controller+stylus methods ($W = 7.5$, $p = 0.015$, $R = 0.773$ ($R \geq 0.5$ is considered a large effect size)) and between the hands and controller+stylus methods ($W = 0.0$, $p = 0.003$, $R = 1.000$). The degrees of freedom was 11 in all pairs, as in all subsequent Wilcoxon signed rank tests. There were no statistically significant differences between the hands and mouse methods ($W = 16.0$, $p = 0.76$).

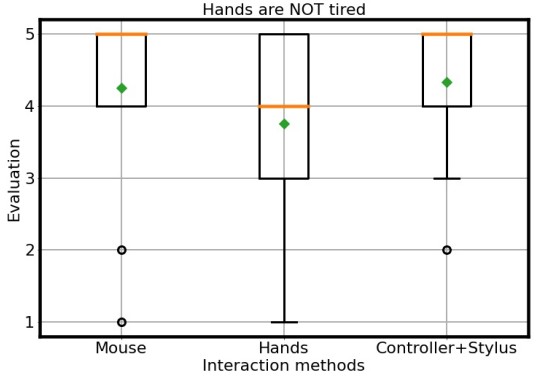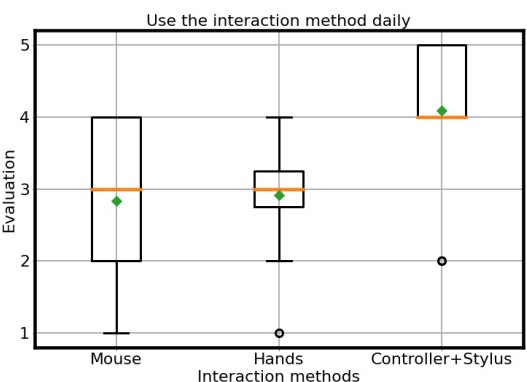

**Figure 5.** Box plot visualizations of the fatigue and daily use evaluations. The evaluation of fatigue (**left**): none of the methods were found to be particularly tiring. The evaluation of possible daily use (**right**): controller+stylus was significantly more usable for daily use than the other methods.

We asked the participants to evaluate both object manipulation and marking separately. In object manipulation evaluation, there were statistically significant differences in naturalness between controller+stylus and mouse ($W = 0.0$, $p = 0.003$, $R = 1.000$) and controller+stylus and hands ($W = 4.0$, $p = 0.009$, $R = 0.879$). There was no statistically significant difference between mouse and hands ($W = 10.0$, $p = 0.13$). In object manipulation easiness, controller+stylus had a statistically significant difference between mouse and hands ($W = 0.0$, $p = 0.003$, $R = 1.000$ for both methods), see Figure 6. There were no statistically significant differences between mouse and hands (the evaluation values happened to be equal for all participants). In manipulation accuracy evaluation, we found a statistically significant difference between the controller+stylus method and hands method ($W = 0.0$, $p = 0.003$, $R = 1.000$). There were no statistically significant differences between mouse and controller+stylus ($W = 2.5$, $p = 0.027$) or hands and mouse ($W = 11.0$, $p = 0.32$). In the object marking evaluation (Figure 7), the only significant difference was measured between the controller+stylus method and mouse method in easiness ($W = 0.0$, $p = 0.009$, $R = 1.000$). There were no statistically significant differences between hands and controller+stylus ($W = 6.5$, $p = 0.053$) or hands and mouse ($W = 20.0$, $p = 0.76$).

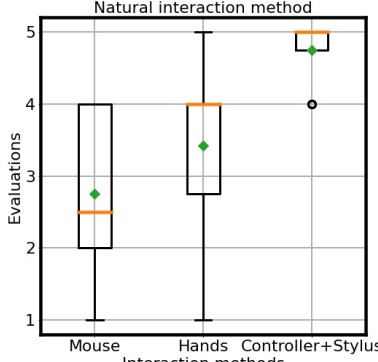
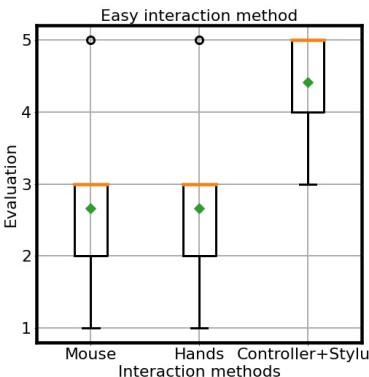
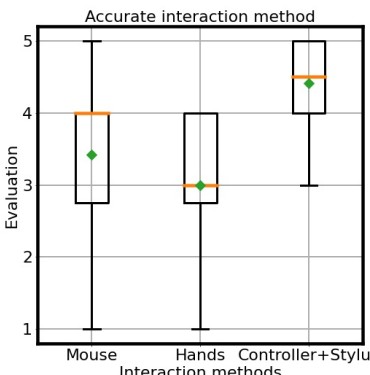

**Figure 6.** Box plot visualizations of the interaction method evaluations. The evaluation of interaction method naturalness (**left**), easiness (**middle**), and accuracy (**right**). Controller+stylus was the most liked method in these features. There were significant differences between controller+stylus and hands in all attributes (naturalness, easiness, and accuracy) and between controller+stylus and mouse in naturalness and easiness.

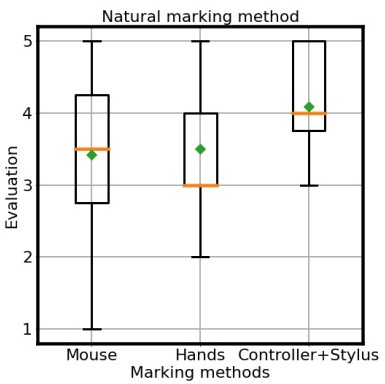
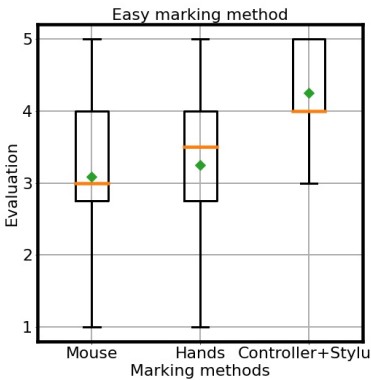
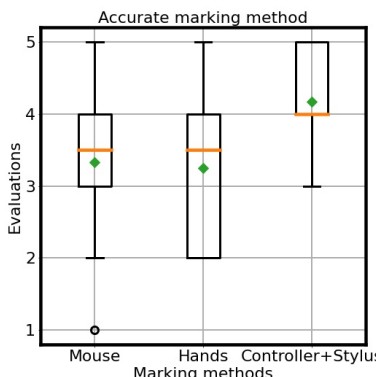

**Figure 7.** Box plot visualizations of the marking method evaluations. The evaluation of marking method naturalness (**left**), easiness (**middle**), and accuracy (**right**). Median values in these features are rather similar, and significant difference was found only in marking easiness between controller+stylus and mouse.

Multiple participants commented that the controller interaction felt stable and that it was easy to move and rotate the 3D object with the controller. The participants also commented that holding a physical device in hand so that its weight could be felt increased the feel of naturalness. Not all comments agreed, and one participant felt the VR stylus was accurate while another participant said it felt clumsy.

When asked, 11 out of 12 participants ranked controller+stylus the most liked method. The distribution of ranking values is shown in Table 1. The ranking values of the controller+stylus method were statistically significantly different to the mouse ($W = 4.5$, $p = 0.008$, $R = 0.885$) and hands ($W = 0.0$, $p = 0.003$, $R = 1.000$). There was no statistically significant difference between mouse and hands ($W = 24.0$, $p = 0.27$).

**Table 1.** The number of mentions of different rankings of the interaction methods when asked for the most liked (1st), the second most liked (2nd), and the least liked (3rd) method.

| Condition | Ranking | | |
|---|---|---|---|
| | 1st | 2nd | 3rd |
| Mouse | 1 | 7 | 4 |
| Hands | 0 | 4 | 8 |
| Controller+Stylus | 11 | 1 | 0 |

## 6. Discussion

In this study, we were looking for the most feasible interaction method in VR for object manipulation and medical marking. The controller+stylus method was overall the most suitable for a task that requires both object manipulation and marking. The controller+stylus method was the most liked in all subjective features, while the mouse and hands conditions were evaluated very similarly. The smallest number of markings was performed with controller+stylus, but no significant differences were found. There were statistically significant differences between the methods in daily use, interaction naturalness, and easiness. Controller+stylus was statistically significantly more accurate in object manipulation than hands ($p = 0.003$), and easier to use than mouse ($p = 0.003$). Without earlier experience with the VR stylus, the participants had difficulties in finding the correct button when marking with the stylus. The physical stylus device cannot be seen when wearing the VR headset and the button could not be felt clearly. Even though the controller+stylus combination was evaluated as natural and the most liked method in this study, the hand-held devices may feel inconvenient [8]. In our study, some participants liked the physical feel of devices. However, our result was based on the subjective opinions of participants, and that might change depending on the use case or devices.

The low hand tracking accuracy is an obvious weakness in the method. There are many possible reasons for that. Hand inaccuracy can be seen in the large number of markings and large distribution in trial completion times with hands as the participants were not satisfied with their first marking. The hands method was the only method where only one participant succeeded with a minimum of five markings, when, by other methods, several participants succeeded in the task with five markings. One explanatory factor can be the lack of hand tracking fidelity that has also been noticed in other studies [8,31]. The vision-based hand tracking system that uses a camera on a HMD does not always recognize the hand gesture well enough and, as a result, the participant must repeat the same gesture or movement multiple times to succeed. This extra work also increases the fatigue in hands. Even though the fatigue was low with all interaction methods, this study did not measure the fatigue of long-term activity. These are clear indications that hands interaction needs further development before it can be used in tasks that need high marking accuracy. Several earlier studies have reported the hands inaccuracy compared to controllers [8,31,33].

Passive haptics were available with the mouse and when marking with the VR stylus. With hands, there was only visual feedback. The lack of any haptic feedback might have affected the marking accuracy as well because the accuracy was much better with the physical stylus. Li et al. [10] found that, with the low marking difficulty, the mouse with a 2D display was faster than the kinesthetic force feedback device in VR. For high marking difficulty, the other VR interface that used a VR controller with vibrotactile feedback was better than the 2D interface. They found that a mouse in a 2D display has fast pointing capability but, in our study, the trial completion times did not vary between the mouse and the other methods. Li et al. described the fact that manipulating the viewing angle is more flexible when wearing a HMD than with a mouse in a 2D display. In VR interfaces, the participant can rotate the 3D object while changing the viewing angle by moving their head. In our study, all methods used HMD, so the change in viewing angle was equally flexible.

The mouse was a statistically significantly more accurate marking method than hands. With the mouse, not being able to see the device during use was not perceived as a problem. The mouse was not affected by some of the issues that were noticed with hands or controller+stylus. There were no sensor fidelity issues with the mouse, and the mouse was a familiar device to all participants. Only the ray that replaced the cursor was an unfamiliar feature and caused some problems. We found that the ray worked well with simple 3D objects but there were a lot of difficulties with complex objects where the viewing angle needed to be exactly right to reach the target. If any part of the 3D object blocked the ray, the target could not be marked. When the target was easy to mark, the accuracy using the mouse was high. It can be stated that the mouse was an accurate method in VR but, for all other measured properties, controller+stylus were measured to be better.

Both the target and the landmark were spheres in a 3D environment. During the study, it was noticed that, when a participant made their marking in the same location as the target, the marking sphere disappeared inside the target sphere. This caused uncertainty if the marking was lost or if it was in the center of the target. This may have affected the results when the participants needed to remark to be able to see the landmark that was not in the center of the target. In future studies, the marking sphere should be designed to be bigger in size than the target and transparent so that the participant can be sure about the location of both spheres.

### 6.1. Limitations

The robustness of the interaction methods varied. The mouse, for example, was very robust against any kind of sensor detection issues. The mouse was also familiar to all participants and, therefore, there were no novelty issues. However, the complexity of the 3D object was challenging in cases for the mouse ray use. The hands method was rather sensitive to the camera position and specific hand pose, and the recognition algorithm obviously could not always see the hand properly, which led to poor tracking. The VR controller+stylus combination was quite robust for the position and orientation detection and the participants found it easy to use.

A limiting factor in generalizing the test results was the characteristics of the participants available. We were using a rather homogeneous group of (rather young) participants that were all somewhat experienced with various interaction devices. Even then, the VR controller and stylus were new to some of the participants.

Our focus was in comparing three different interaction and marking methods and their suitability for the medical marking task. To simplify the experimental setup, the experiment was conducted with simplified medical images, which may have led to overly optimistic results for the viability of the methods. Even then, there were some problems with the mouse interaction method.

### 6.2. Future Research

To confirm the results for more realistic content, a similar study should be conducted in future with authentic material utilizing, for example, original CBCT images in VR instead of the simplified ones. In the future, we expect that, as sensor technology improves and the hands condition becomes more functional, a similar experiment may give different results.

Future research may also investigate multimodal interaction methods to support even more natural alternatives. Speech is the primary mode for human communication [40]. Suresh et al. [41] used three voice commands to control gestures of a robotic arm in VR. Voice is a suitable input method in cases where hands and eyes are continuously busy [33]. Pfeuffer et al. [42] studied gaze as an interaction method together with hand gestures but found that both hand and gaze tracking still lack tracking fidelity. More work is still needed, as Nukarinen et al. [43] stated that human factor issues made the gaze the least preferred input method in an object selection task in VR.

## 7. Conclusions

Three-dimensional medical images can be manipulated in VR environments for medical surgery planning. During the planning process, one needs to interact with the 3D objects and be able to make markings of high accuracy on them. We evaluated the feasibility of three different VR interaction methods: a mouse, hands, and a controller+stylus combination in virtual reality. Based on the results, we can say that the Valve Index controller and Logitech VR Ink stylus combination was the most feasible for tasks that require both 3D object manipulation and high marking accuracy in VR. This combination did not have issues with complex 3D objects and sensor fidelity was better than with hands interaction. Statistically significant differences were found between the controller combination and the other methods.

Hand-based interaction was the least feasible for this kind of use. The hands and mouse methods were evaluated almost equal in feasibility by participants. Free hands usage cannot be proposed for accurate marking tasks. Mouse interaction was more accurate than controller+stylus. In detailed tasks, the mouse could replace the free hands interaction. The discrepancy between the 2D mouse and the 3D environment needs to be solved before the mouse could be considered a viable interaction method in VR.

The results show that the users approve the easiness and fluency of the use of the controller+stylus, because of the tracking accuracy and natural interactions, such as being able to grab the object and point at it with the stylus. As sensing technologies are improving, in the future, we expect that new interaction methods will reach the same level of "naturalness" and become acceptable.

**Author Contributions:** Conceptualization, H.-R.R., J.K., H.M., R.R., and S.K.K.; methodology, H.-R.R. and J.K.; software, S.K.K.; validation, H.-R.R., J.K., and S.K.K.; formal analysis, H.-R.R.; investigation, H.-R.R. and J.K.; resources, R.R.; data curation, H.-R.R. and J.K.; writing—original draft preparation, H.-R.R. and J.K.; writing—review and editing, H.-R.R., J.K., S.K.K., H.M., J.J., and R.R.; visualization, J.K.; supervision, R.R.; project administration, R.R.; funding acquisition, R.R. and J.J. All authors have read and agreed to the published version of the manuscript.

**Funding:** This work has been funded by Business Finland, project Digital and Physical Immersion in Radiology and Surgery (decision number 930/31/2019) and by the Academy of Finland, project Explainable AI Technologies for Segmenting 3D Imaging Data (decision number 345448).

**Institutional Review Board Statement:** This study was conducted according to the guidelines of the Declaration of Helsinki. Ethical review and approval were waived for this study, due to the nature of the experimental tasks and the participant population.

**Informed Consent Statement:** Informed consent was obtained from all subjects involved in the study.

**Data Availability Statement:** The data are available upon request.

**Acknowledgments:** The authors wish to thank the dentomaxillofacial radiologists that provided the knowledge of their specialty.

**Conflicts of Interest:** The authors declare no conflict of interest.

## Abbreviations

The following abbreviations are used in this manuscript:

| | |
|---|---|
| 2D | two-dimensional |
| 3D | three-dimensional |
| CBCT | cone beam computed tomography |
| HMD | head-mounted display |
| VR | virtual reality |

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
