# Peer review of "Comparison of a VR Stylus with a Controller, Hand Tracking, and a Mouse for Object Manipulation and Medical Marking Tasks in Virtual Reality"

_applsci, doi:10.3390/app13042251_

Round 1
Reviewer 1 Report
The paper presents medical surgery planning, virtual reality (VR) and provides a working environment, where 3D images of the operation area can be utilized. The aim of paper is to study the viability of the methods in VR for object manipulation and marking tasks conducted in medical surgery planning. The paper is well-written and presents interesting results. However, the authors are requested to address and revise it according to the suggestions listed below:
· 1. The main contributions of this manuscript must be summarized in the Introduction.
· 2. The authors have not identified the novelty nor described how VR Stylus with a Controller, Hand Tracking and a Mouse for Object Manipulation and Medical Marking Tasks in Virtual Reality is different from existing work.
· 3. The discussion of the results needs to include the strengths and weaknesses of VR Stylus with a Controller, Hand Tracking and a Mouse for Object Manipulation and Medical Marking Tasks in Virtual Reality. The authors should clarify the pros and cons of the methods. What are the limitation(s) methodology(ies) adopted in this work? Please indicate practical advantages, and discuss research limitations. These limitations can be organized around simple distinctions of the choices you made in your study regarding who, what, where, when, why, and how. To have an unbiased view in the paper, there should be some discussions on the limitations of the compared methods.
· 4. Statistical tests to judge about the significance of the method’s results is absent. Without such a statistical test, the conclusion cannot be supported. It will be good to present a statistical test in the comparison of the results with other published methods. This can help to support the claim on improved results obtained with the selection methods studied.
· 5.The robustness about the method has not been discussed. Parameter or sensitivity analysis has not been performed.
· 6. Some more recommendations and conclusions should be discussed about the paper considering the experimental results. The Conclusion section is weak. Furthermore, there is not any discussion section about the results. The conclusion section needs significant revisions. It should briefly describe the findings of the study and some more directions for further research. The authors should describe academic implications, major findings, shortcomings, and directions for future research in the conclusion section. The conclusion in its current for is confused in general. Concerning Conclusion section, it would be better "Conclusions and Future Research", and it is strongly suggested to include future research of this manuscript. What will be happen next? What we supposed to expect from the future papers?
· 7. There are too many spelling and grammar mistakes in the paper. It needs proper spelling and grammar checking.
Reviewer 2 Report
This paper tackles an interesting topic since VR and AR have the potential to be employed in a variety of different settings. Although the paper has a good theoretical background, there are several concerns related to the study. First, the authors have not provided evidence that they examined requirements for statistical tests they examined. Second, when presenting results of the T, Friedman, and Wilcoxon test, the authors should report both the result of a particular test (e.g. t, Z... value) together with degrees of freedom, regardless of whether the result is significant or not. Third, I am not convinced that based on only 12 study participants any sound inferential statistics can be conducted and a conclusion be drawn. Fourth, in the discussion, reported findings should be compared with current relevant studies and the authors should emphasize the contribution of their work to the extant body of knowledge. Fifth, the authors should clarify the implications of the reported findings and limitations of the study they carried out. Finally, the figures should be shown after they have been mentioned in the text. Considering all the above, the authors are encouraged to rework their paper.
Reviewer 3 Report
This study compared a mouse, hand tracking, and a combinationof a VR stylus and a grab-enabled VR controller as interaction methods in VR.
The stylus and controller combination was the most preferred interaction method.
However, there are still many problems in this manuscript as follows:
1. Please clarify the novelty of this manuscript.
2. Are there other interaction methods besides the three in this manuscript? If so, please explain the advantages and disadvantages of different methods and why these three interaction methods were chosen.
3. In addition to the several problem statements mentioned on page 7, are there other indicators that positively impact the analysis of experimental results?
4. The rotation and translation were mentioned in the manuscript, whether to keep the rotation angle and displacement as equal as possible in the experiment.
Round 2
Reviewer 1 Report
Dear Editor,
The authors have incorporated all suggestions in the manuscript. Therefore, I accept it for publication in this journal after minor English and Grammatical check.
Regards,
Dr. Ghani Ur Rehman